# Metacontrol of decision-making strategies in human aging

**Florian Bolenz[1]\*, Wouter Kool[2,3], Andrea MF Reiter[1,4,5], Ben Eppinger[1,6,7]**

[1]Chair of Lifespan Developmental Neuroscience, Faculty of Psychology, Technische Universität Dresden, Dresden, Germany; [2]Department of Psychology, Harvard University, Cambridge, United States; [3]Department of Psychological & Brain Sciences, Washington University in St. Louis, St. Louis, United States; [4]Wellcome Centre for Human Neuroimaging, University College London, London, United Kingdom; [5]Max Planck UCL Centre for Computational Psychiatry and Ageing Research, London, United Kingdom; [6]Department of Psychology, Concordia University, Montreal, Canada; [7]PERFORM centre, Concordia University, Montreal, Canada

**Abstract** Humans employ different strategies when making decisions. Previous research has reported reduced reliance on model-based strategies with aging, but it remains unclear whether this is due to cognitive or motivational factors. Moreover, it is not clear how aging affects the metacontrol of decision making, that is the dynamic adaptation of decision-making strategies to varying situational demands. In this cross-sectional study, we tested younger and older adults in a sequential decision-making task that dissociates model-free and model-based strategies. In contrast to previous research, model-based strategies led to higher payoffs. Moreover, we manipulated the costs and benefits of model-based strategies by varying reward magnitude and the stability of the task structure. Compared to younger adults, older adults showed reduced model-based decision making and less adaptation of decision-making strategies. Our findings suggest that aging affects the metacontrol of decision-making strategies and that reduced model-based strategies in older adults are due to limited cognitive abilities.

DOI: https://doi.org/10.7554/eLife.49154.001

**\*For correspondence:**
florian.bolenz@tu-dresden.de

**Competing interests:** The authors declare that no competing interests exist.

## Introduction

In order to cope efficiently with environmental demands, humans as well as other biological and artificial agents have to engage in metacontrol, that is they constantly have to decide how to allocate their limited computational resources (*Boureau et al., 2015*; *Gershman et al., 2015*; *Griffiths et al., 2019*). Recent research in psychology and neuroscience has advanced our understanding of metacontrol in humans (e.g., *Kurzban et al., 2013*; *Shenhav et al., 2017*), however it is still unclear how metacontrol changes across the lifespan. In our current study, we investigate how aging affects the metacontrol of decision-making strategies. For this purpose, we focus on *model-free* and *model-based decision making* (*Daw et al., 2011*; *Dolan and Dayan, 2013*) – two strategies that are reflective of a broader distinction between automatic and deliberative modes of information processing as put forward by prominent dual-system theories (*Balleine and O'Doherty, 2010*; *Kahneman, 2003*; *Sloman, 1996*).

Within the computational framework of reinforcement learning (*Sutton and Barto, 1998*), model-free decision making represents the strategy to repeat actions that previously led to desirable outcomes and to avoid actions that previously led to undesirable outcomes, corresponding to Thorndike's law of effect (*Thorndike, 1911*). Model-free control only requires a single value update and therefore it is computationally efficient and does not rely strongly on cognitive resources. However,

it can be inflexible and inaccurate because it requires direct experience with the consequences of an action in order to update the corresponding reward expectation.

Another strategy for decision making is to store an internal model about contingencies in the environment (e.g., how taking an action leads from one state to another and how rewards are distributed across states) and to use this model to predict the consequences of available actions (*Tolman, 1948*). Computationally, this strategy has been formalized as model-based decision making. This strategy can account for sudden changes in the environment more quickly and may be more accurate in complex environments; however, it comes with the cost of higher cognitive demands.

It is commonly assumed that model-free and model-based strategies involve different cognitive mechanisms and compete for behavioral control (*Daw et al., 2005*; *Kool et al., 2017a*). Indeed, humans show signs of both strategies during decision making (*Daw et al., 2011*; *Gläscher et al., 2010*), and they dynamically regulate the balance between model-free and model-based control on a moment-to-moment basis. It has been suggested that this metacontrol of decision-making strategies is based on a cost-benefit analysis (*Kool et al., 2017b*; *Kool et al., 2018*). This framework states that people weigh the cognitive costs of each decision-making strategy against its potential benefits in order to decide how to invest mental resources. For example, people increase model-based control when rewards are amplified (*Kool et al., 2017b*). Conversely, model-based control decreases when the task becomes more complex and thus model-based planning gets more effortful (*Kool et al., 2018*).

Older adults show reduced model-based control during decision making compared to younger adults (*Eppinger et al., 2013*). This finding could be explained by aging-related difficulties in representing the structure of the experimental task (i.e., the contingencies between actions and states in the task) which would limit the use of a task model to accurately predict future outcomes (*Eppinger et al., 2015*; *Hämmerer et al., 2019*). On the neurobiological level, difficulties in the representation of the task structure could result from structural decline of prefrontal and hippocampal regions (*Raz et al., 2005*; *Resnick et al., 2003*) that have been suggested to be involved in the encoding of mental task models (*Koechlin, 2016*; *Schuck et al., 2016*; *Vikbladh et al., 2019*).

However, the observed reduction of model-based control in older adults does not necessarily have to be an impairment: The sequential decision-making task that is most frequently used to determine individual reliance on model-free versus model-based control (*Daw et al., 2011*) lacks a control-reward trade-off (*Kool et al., 2016*). That is, greater model-based control does not lead to higher pay-offs in the task compared to model-free control. When considering the greater cognitive costs that come with a model-based strategy, it may be optimal to engage in a model-free strategy, particularly for individuals with limitations in cognitive abilities such as older adults.

In the present study, we aimed to distinguish between these explanations for the finding that older adults show reduced model-based control. Specifically, we were interested in whether older adults continue to show reduced model-based control even if this strategy leads to better pay-offs than model-free decision making. For this purpose, we make use of a recent version of the sequential decision-making task that has a control-reward trade-off (*Kool et al., 2016*). If aging-related changes in model-based control are due to difficulties in representing the task structure, older adults should show reduced model-based control in this modified task. Alternatively, if the aging-related changes reported previously (*Eppinger et al., 2013*) are caused by a reduced willingness to engage in a more effortful strategy without a related improvement in terms of rewards, reliance on model-based control should be aligned for both age groups when an association between model-based control and reward is introduced. Furthermore, previous studies have found that surprising transitions elicit a slowing of response times (*Decker et al., 2016*; *Deserno et al., 2015*; *Shahar et al., 2019*). If task structure representations become less accurate with aging, this should also be reflected in less differentiated responses to surprising and unsurprising transitions.

In order to investigate how aging affects the metacontrol of decision-making strategies, we compared the dynamic regulation of decision-making strategies between younger and older adults. To this end, we manipulated the magnitude of rewards across trials (*Kool et al., 2017b*). If older adults take into account differences in the benefits of model-based decision making just like younger adults do (*Kool et al., 2017b*), they should increase model-based control when rewards are amplified. Alternatively, if metacontrol is reduced in older adults, they should show less adaptation of model-based control to different reward magnitudes. Additionally, we added an experimental condition to the existing paradigm in which the task transition structure changes repeatedly. In this condition,

participants constantly have to update their internal representation of the task which should inflict higher costs on model-based control. If younger and older adults take into account differences in the costs of model-based decision making, both age groups should decrease model-based control when the task transition structure becomes more variable. Alternatively, less adaptation of model-based control to the variability of the task structure in older adults would point to reduced meta-control with aging.

## Results

We analyzed the behavior of 62 younger adults (age range: 18–30 years, mean age: 22.7 years) and 62 older adults (age range: 57–80 years, mean age: 70.2 years) in a sequential decision-making task that dissociates model-free and model-based decision making. Importantly, in this task higher model-based decision making leads to better outcomes in terms of monetary reward (*Kool et al., 2016*). In brief, every trial started randomly in one of two first-stage states that offered a choice between two spaceships (*Figure 1A*). This choice determined which second-stage state, a red or a purple planet, would be encountered. For each first-stage state, one spaceship would lead to the red planet, whereas the other would lead to the purple planet. Each planet afforded the opportunity to earn reward, represented by a certain number of pieces of 'space treasure'. The amount of space treasure was independent for each planet and drifted over the course of the experiment so that participants had to constantly update their reward expectations.

This task is able to dissociate model-free from model-based decision making. To see this, note that each first-stage state offers an identical choice between transitioning to the red and the purple planet. A pure model-based decision maker is able to use this equivalence to transfer experiences learned in one starting state to the other, because it plans over the transition structure toward the second-stage goals. For example, an unexpectedly good outcome on the previous trial should affect choices on the next trial, regardless of whether it starts in the same starting state, because the agent assigns value to the spaceships by combining the transition structure with the second-stage reward expectations. A pure model-free decision maker does not consult the transition structure of the task, but instead relies on action-reward contingencies. Therefore, reward outcomes on the previous trial only affect choices on the current trial if the same actions are available. In other words, a model-free agent is unable to transfer experiences obtained after one pair of spaceships to the other pair (*Doll et al., 2015*). Here, we use this distinction between these strategies in order to test our hypotheses.

To vary the cost and benefits associated with model-based decision making in the two age groups, we introduced two factorial manipulations to this task: a stakes manipulation and a transition stability manipulation. First, similar to previous studies (*Kool et al., 2017b*; *Patzelt et al., 2019*), every trial started with a randomly picked cue that indicated whether it was a *low-stakes* or a *high-stakes* trial (*Figure 1B*). In low-stakes trials, participants obtained one point for every piece of space treasure that they received and in high-stakes trials, participants obtained five points for every piece of space treasure. Second, we manipulated the stability of the task transition structure (i.e., the mapping of spaceships to planets) in blocks of 80 trials (*Figure 1C*): Throughout *stable-transitions blocks*, all spaceships maintained the same destination planet. In *variable-transitions blocks*, every 6 to 14 trials, the pair of spaceships in one of the first-stage states swapped their destination planets. These reversals of the transition structure required participants repeatedly to update their internal model of the task and thus increased the demands on model-based decision making. At the beginning of each trial block, we informed participants about the transition stability condition for the upcoming trials.

We compared the overall baseline-corrected reward obtained between age groups (*Figure 2A*). An effect-coded hierarchical regression analysis revealed an age-difference in reward ($\bar{\beta}_{\text{age group}} = 0.25$, 95% credible interval (CI) = [0.17, 0.33]), indicating that younger adults performed on average better than older adults.

### Reinforcement-learning modeling

We used a hybrid reinforcement-learning model to assess the individual reliance on model-free versus model-based decision making. In this reinforcement-learning model, a model-free learner learns independent reward expectations for each of the four spaceships in the first-stage states as well as

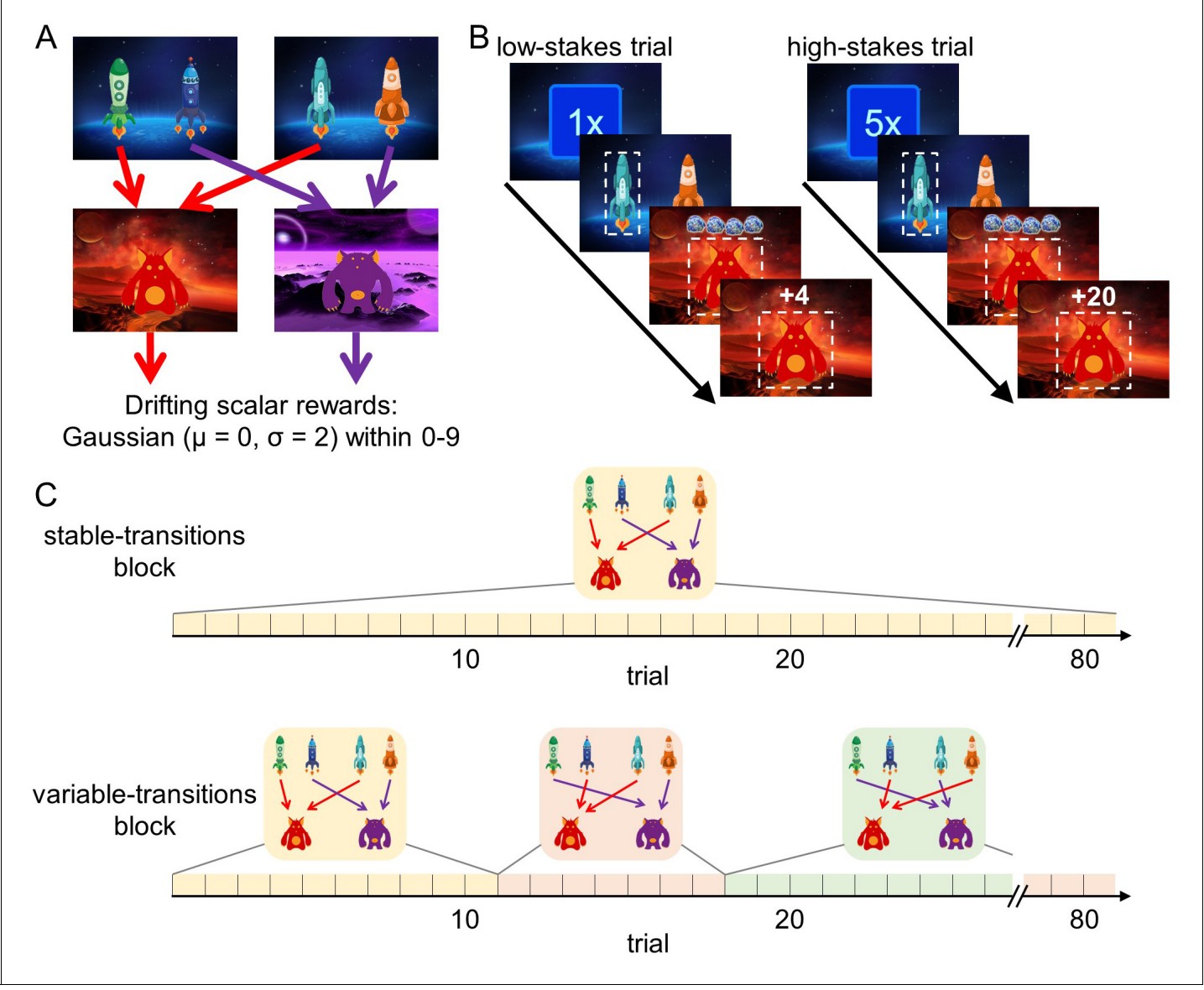

**Figure 1.** Design of the sequential decision-making task. (**A**) State transition structure of the task. (**B**) Trial structure. At the beginning of each trial, a stakes cue signaled the stakes condition of the current trial (low-stakes trial vs. high-stakes trials). Participants started in one of two first-stage states where they selected one spaceship. They then transitioned to a planet (second stage) where they received a certain amount of space treasure. Space treasure was converted into points depending on the stakes condition. (**C**) In stable-transitions blocks, spaceships maintained their destination planets throughout a block of 80 trials. In variable-transitions blocks, every 6 to 14 trials, one pair of spaceships swapped their destination planets.
DOI: https://doi.org/10.7554/eLife.49154.002

for each of the second-stage planets and updates these reward expectations based on reward prediction errors. In contrast, a model-based learner learns a transition structure that represents to which planet the choice of a spaceship will lead. By means of this knowledge about the task structure, model-based decision making can use the reward expectations associated with the two planets when selecting a spaceship.

In order to capture individual differences in the reliance on these two strategies, the hybrid reinforcement-learning model includes a *model-based weight* that determines their relative contribution. Model-based weights close to 0 reflect model-free decision making and model-based weights close to one reflect model-based decision making. Intermediate values of the model-based weight indicate that behavior reflects a combination of both decision-making strategies. When fitting the

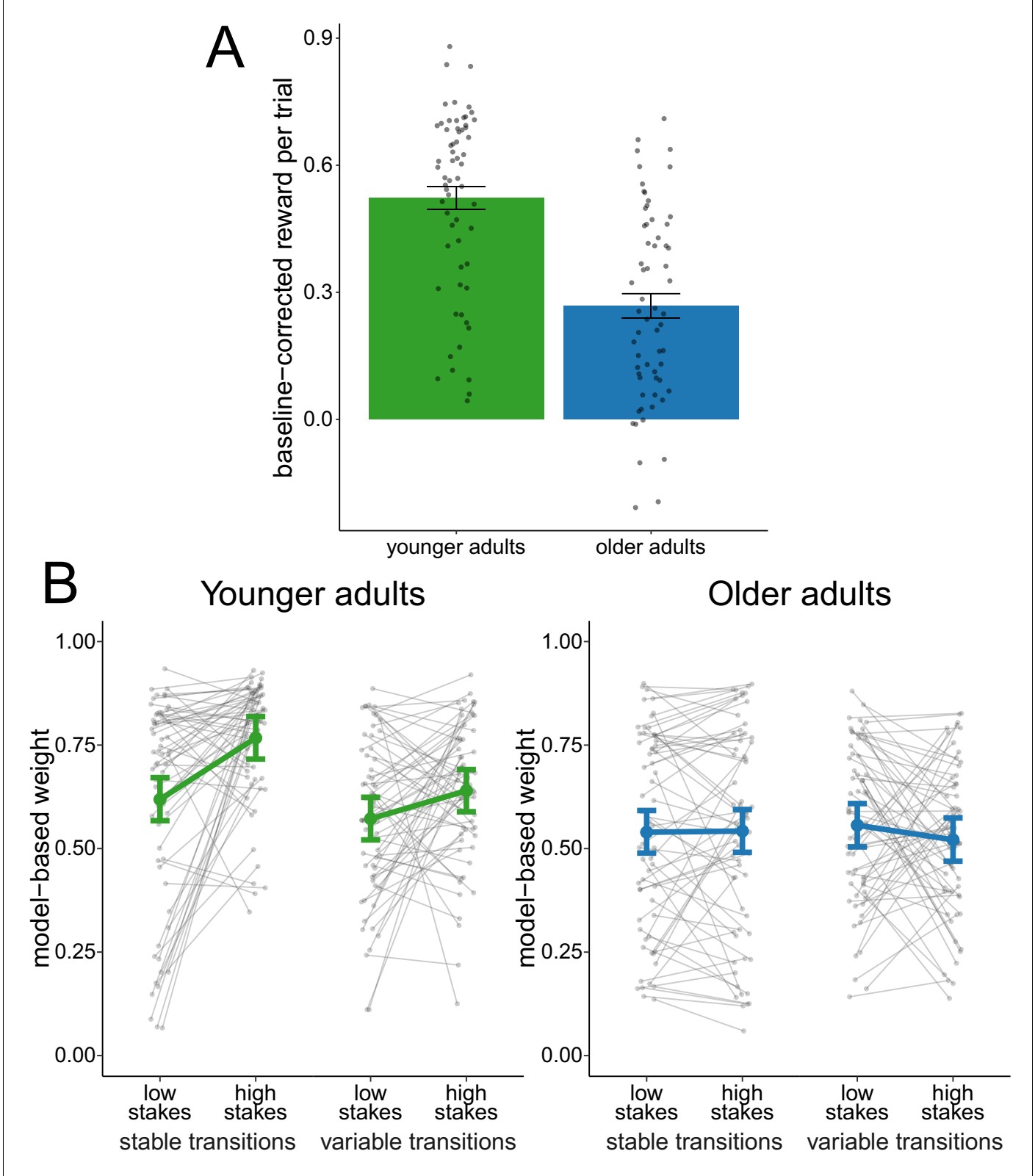

**Figure 2.** Analysis of complete sample. (**A**) Reward per trial (baseline-corrected). Gray dots indicate values for individual participants and bars indicate group means with error bars representing standard error of the mean. (**B**) Degree of model-based control. Model-based weights are depicted for the complete sample of younger and older adults as a function of stakes condition and transition variability condition. Gray dots indicate values for

*Figure 2 continued on next page*

*Figure 2 continued*

individual participants. Colored dots indicate group means as predicted by the hierarchical regression model with error bars representing Bayesian 95% credible intervals.

DOI: https://doi.org/10.7554/eLife.49154.003

The following figure supplement is available for figure 2:

**Figure supplement 1.** Degree of model-based control under the assumption of perfect transition learning.

DOI: https://doi.org/10.7554/eLife.49154.004

model, the model-based weight was estimated as a free parameter. In order to test for differences in decision-making strategies between task conditions, we estimated for every participant four model-based weights: one for each combination of stakes condition and transition stability condition (see Materials and methods section for details).

We compared model-based weights (*Figure 2B*) with an effect-coded hierarchical regression analysis. This analysis revealed evidence for a main effect of age group ($\bar{\beta}_{\text{age group}}$ = 0.11, 95% CI = [0.05, 0.16]), indicating that older adults employed reduced model-based control compared to younger adults. This effect replicates previous findings (*Eppinger et al., 2013*). We found evidence for a main effect of stakes condition ($\bar{\beta}_{\text{stakes}}$ = 0.05, 95% CI = [0.01, 0.08]), indicating increased model-based control in high-stakes trials. The analysis also revealed a main effect of transition stability condition ($\bar{\beta}_{\text{transition}}$ = 0.04, 95% CI = [0.01, 0.07]), indicating that model-based control was lower during variable-transitions blocks compared to stable-transitions blocks. Finally, we obtained evidence for an interaction effect of age group and stakes condition ($\bar{\beta}_{\text{age group} \times \text{stakes}}$ = 0.12, 95% CI = [0.06, 0.18]) as well as and interaction effect of age group and transition stability condition ($\bar{\beta}_{\text{age group} \times \text{transition}}$ = 0.09, 95% CI = [0.03, 0.15]), which indicates that the effects of both stakes and transition stability condition were reduced in older adults.

In the findings reported above, we only allowed the model-based weight to vary between experimental conditions. Thus, any potential changes in processes other than the reliance on different decision-making strategies would be forced on this parameter. To account for this potential confound we also fitted an exhaustive version of the reinforcement-learning model that varied all model parameters between experimental conditions (cf. Table S1 in *Supplementary file 1*). When analyzing differences in the model-based weights in this exhaustive model, we found a main effect of age group ($\bar{\beta}_{\text{age group}}$ = 0.12, 95% credible interval (CI) = [0.07, 0.17]) and a main effect of stakes condition ($\bar{\beta}_{\text{stakes}}$ = 0.04, 95% CI = [0.01, 0.07]), replicating the results from the standard model. Furthermore, consistent with the results from the standard model, there was an interaction effect of age group and stakes condition ($\bar{\beta}_{\text{age group} \times \text{stakes}}$ = 0.07, 95% CI = [0.01, 0.14]) as well as and interaction effect of age group and transition stability condition ($\bar{\beta}_{\text{age group} \times \text{transition}}$ = 0.12, 95% CI = [0.06, 0.19]). Thus, this analysis shows that the observed differences in model-based weights cannot be completely accounted for by decision processes other than model-based control.

Additionally, we found reduced parameter estimates for the transition learning rate in older adults ($\bar{\beta}_{\text{age group}}$ = 0.22, 95% CI = [0.14, 0.31]). The transition learning rate indicates how quickly observations about changes in the task transition structure are incorporated into the mental representation of the task (but see Materials and methods section for an alternative interpretation). The lower transition learning rates suggest that older adults have less accurate representations of the task transition structure than younger adults. Further parameters of the standard version of the reinforcement-learning model are reported in Table S2 in *Supplementary file 1*.

To sum up, these findings show reduced reliance on model-based decision making in older adults even in a task where model-based decision making leads to better outcomes. Beyond this, older adults also showed reduced metacontrol of decision-making strategies, that is less dynamic adaptation of decision-making strategies when their associated costs and benefits change.

## Performance-matched sample

To investigate whether the reduced adaptation of model-based decision making was due to differences in task performance between age groups, we analyzed the metacontrol of decision-making strategies in a performance-matched subsample of participants. The matching was achieved by

selecting the participants of both age groups that were the closest in terms of their mean baseline-corrected reward. This procedure yielded a subsample of 26 younger adults and 26 older adults (see *Figure 3A* for the distribution of reward in this subsample). In this performance-matched subsample, a hierarchical regression analysis of model-based weights (*Figure 3B*) showed evidence for a main effect of stakes condition ($\bar{\beta}_{\text{stakes}}$ = 0.07, 95% CI = [0.01, 0.12]) and for an interaction effect of age group and stakes condition ($\bar{\beta}_{\text{age group} \times \text{stakes}}$ = 0.17, 95% CI = [0.07, 0.28]). This indicates a reduced effect of stakes condition in older adults. There was no conclusive evidence for any other main effect or interaction effect.

This matching procedure comes with the downsides of reduced statistical power and potentially biased samples. To control for these potential confounds, we analyzed model-based weights in the total sample with a hierarchical regression analysis while including mean baseline-corrected reward (task performance) and all its interaction terms with stakes condition and transition stability condition as covariates. Consistent with the results in the performance-matched subsample, we observed a main effect of stakes condition ($\bar{\beta}_{\text{stakes}}$ = 0.13, 95% CI = [0.07, 0.19]) and an interaction effect of age group and stakes condition ($\bar{\beta}_{\text{age group} \times \text{stakes}}$ = 0.18, 95% CI = [0.11, 0.25]), which reflects the reduced effect of stakes condition in older adults.

Together, these analyses replicate the finding of reduced metacontrol in older compared to younger adults while controlling for differences in overall task performance between age groups.

## Age differences in transition revaluations

To further examine how the changes in the task transition structure affected choice behavior in the two age groups, we compared trials in which participants observed a change in the task transition structure for the first time (*revaluation trials*) with all other trials from variable-transitions blocks (*non-revaluation trials*). In line with previous results of studies using similar sequential decision-making tasks, we expected slowed reactions immediately after surprising transitions (*Deserno et al., 2015*; *Decker et al., 2016*; *Shahar et al., 2019*), that is for second-stage responses in revaluation trials. A dummy-coded hierarchical regression analysis of log-transformed reaction times at the second stage of the task (*Figure 4A*) revealed a positive slope for older adults ($\bar{\beta}_{\text{old}}$ = 0.32, 95% CI = [0.26, 0.39]), reflecting slowed reaction times for older adults compared to younger adults in non-revaluation trials, and a positive slope for revaluation trials ($\bar{\beta}_{\text{revaluation}}$ = 0.15, 95% CI = [0.11, 0.18]), reflecting that younger adults slowed reaction times in revaluation trials compared to non-revaluation trials. Moreover, we found evidence for an interaction effect of revaluation trial and age group ($\bar{\beta}_{\text{old} \times \text{revaluation}}$ = -0.09, 95% CI = [-0.14, -0.05]). This interaction indicates that younger adults showed a greater slowing of reaction times on revaluation trials compared to non-revaluation trials than older adults and suggests that younger adults were more sensitive to surprising changes of the task structure than older adults.

To control whether slowed reaction times in revaluation trials could be explained by reduced response vigor due to lower reward expectations (*Deserno et al., 2015*), we derived trial-wise estimates for second-stage reward expectations from the reinforcement-learning model and included them as a predictor for log-transformed reaction times in the regression analysis. We still found a positive slope for revaluation trials ($\bar{\beta}_{\text{revaluation}}$ = 0.15, 95% CI = [0.11, 0.18]) which was reduced for older adults ($\bar{\beta}_{\text{old} \times \text{revaluation}}$ = -0.09, 95% CI = [-0.14, -0.05]). The effect of reward expectations was comparably small ($\bar{\beta}_{\text{expecation}}$ = -0.01, 95% CI = [-0.02, -0.00]) and there was no evidence that it was affected by age ($\bar{\beta}_{\text{old} \times \text{expecation}}$ = 0.01, 95% CI = [-0.01, -0.02]). Thus, slowed reaction times in revaluation trials cannot be explained by lower reward expectations in these trials.

The degree of model-based control modulated the slowing of reaction times in revaluation trials (*Figure 4B*). A hierarchical regression analysis of log-transformed reaction times showed an interaction of (low-stakes, variable-transitions) model-based weights and revaluation trials ($\bar{\beta}_{\text{model-based} \times \text{revaluation}}$ = 0.17, 95% CI = [0.04, 0.30]). This interaction indicates that participants relied more on model-based decision making the more their reactions were slowed in revaluation trials. When running the hierarchical regression analysis separately for both age groups, this interaction was replicated for older adults ($\bar{\beta}_{\text{model-based} \times \text{revaluation}}$ = 0.17, 95% CI = [0.01, 0.33]) and with slightly weaker evidence for younger adults ($\bar{\beta}_{\text{model-based} \times \text{revaluation}}$ = 0.14, 95% CI = [-0.05, 0.33], with 93% of the posterior mass above zero). For younger adults, this interaction is consistent with previous findings (*Deserno et al., 2015*; *Shahar et al., 2019*; *Decker et al., 2016*). When performing the same

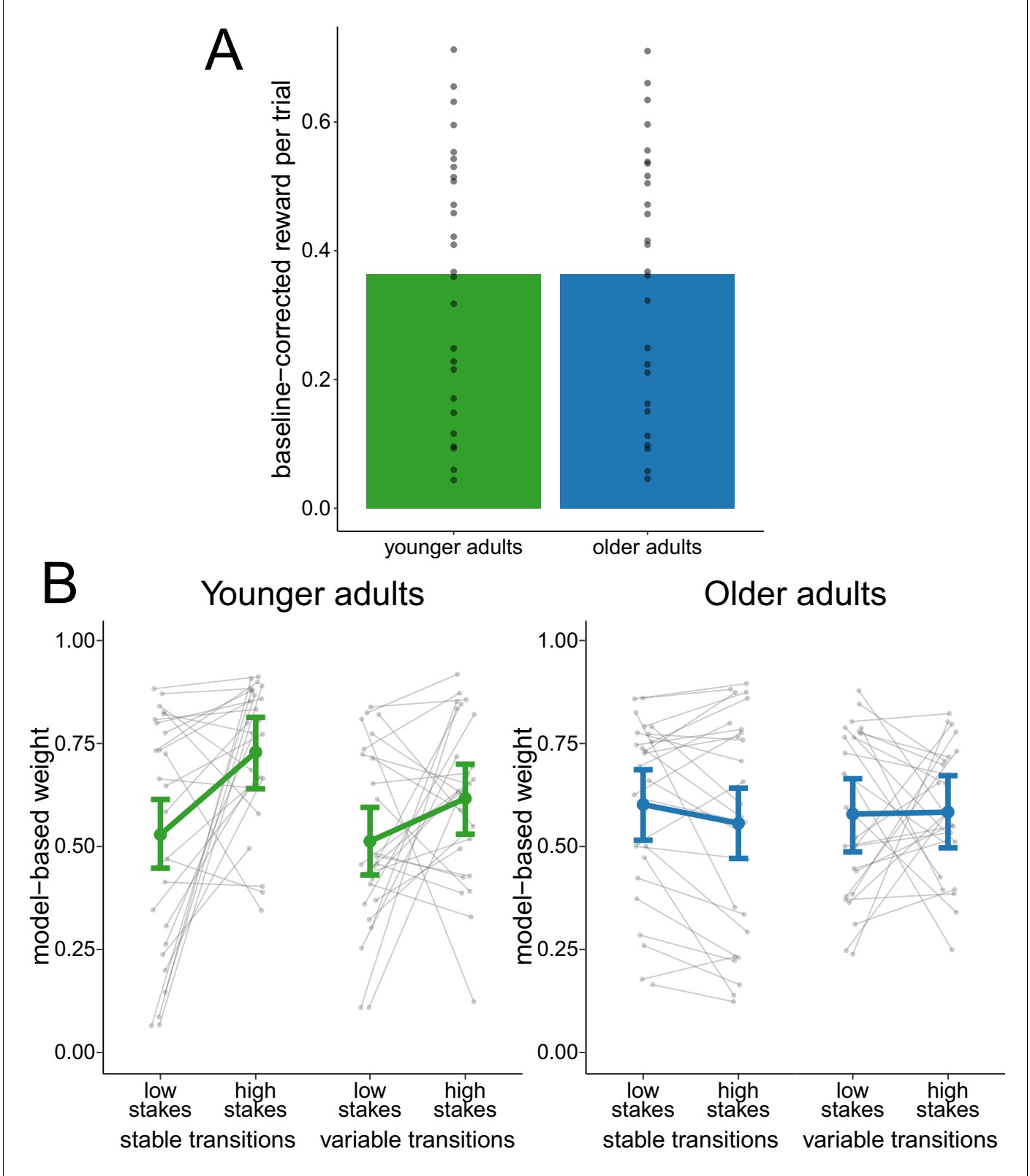

**Figure 3.** Analysis of performance-matched subsample. (**A**) Reward per trial (baseline-corrected) for participants in the performance-matched sample. Gray dots indicate values for individual participants and bars indicate group means. (**B**) Degree of model-based control for younger and older adults in the performance-matched sample as a function of stakes condition and transition variability condition. Gray dots indicate values for individual

*Figure 3 continued*

participants. Colored dots indicate group means as predicted by the hierarchical regression model with error bars representing Bayesian 95% credible intervals.

DOI: https://doi.org/10.7554/eLife.49154.005

analysis with model-based weights from high-stakes trials in variable-transitions blocks, we found similar interactions of model-based weights and revaluation trials although with weaker evidence for this interaction when analyzing both age groups separately (complete sample: $\bar{\beta}_{\text{model-based} \times \text{revaluation}}$ = 0.20, 95% CI = [0.07, 0.34]); older adults: $\bar{\beta}_{\text{model-based} \times \text{revaluation}}$ = 0.16, 95% CI = [-0.02, 0.33]), with 96% of the posterior probability mass above zero; younger adults: $\bar{\beta}_{\text{model-based} \times \text{revaluation}}$ = 0.11, 95% CI = [-0.10, 0.33], with 83% of the posterior probability mass above zero).

This association between model-based decision making and reaction slowing suggests that individual differences in the reliance on model-based decision making might be related to differences in the accuracy of the task structure representation.

## Discussion

Previous findings suggest that older adults show reduced model-based control compared to younger adults (*Eppinger et al., 2013*). However, these findings are based on a task in which model-

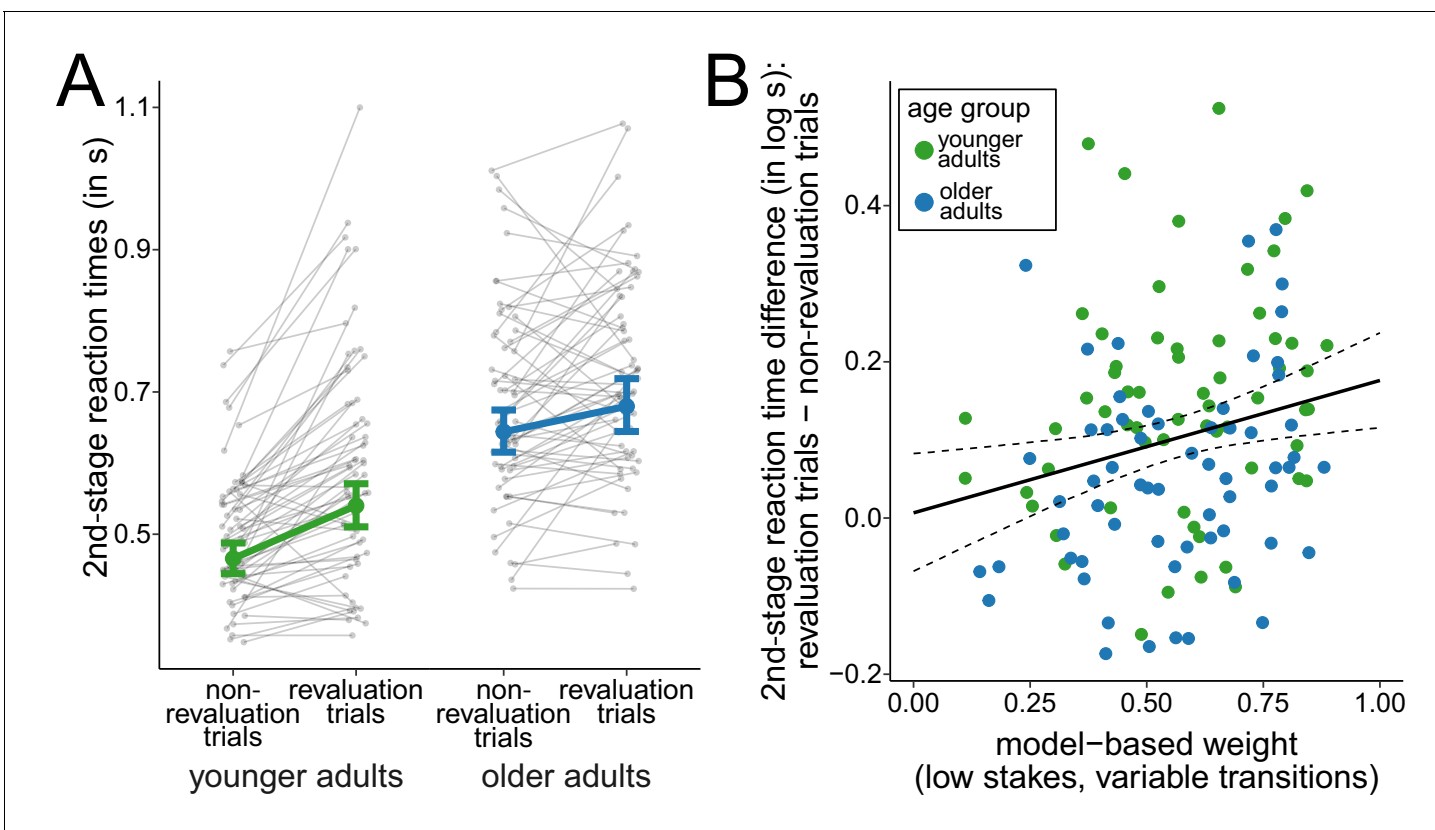

**Figure 4.** Analysis of second-stage reaction times. (**A**) Reaction times at the second stage during variable-transitions blocks for trials in which a change of the task transition structure was observed for the first time (revaluation trials) and all other trials (non-revaluation trials). Gray dots indicate mean values for individual participants. Colored dots indicate group means as predicted by the hierarchical regression model with error bars representing Bayesian 95% credible intervals. (**B**) Relationship between model-based control and reaction time slowing in revaluation trials. Solid line represents prediction from the hierarchical regression model and dashed lines indicate 95% confidence region.

DOI: https://doi.org/10.7554/eLife.49154.006

based control does not lead to more reward (*Kool et al., 2016*). Therefore, it remains unclear whether reduced model-based control with aging is due to limited cognitive capacities or a reduced willingness to engage in a more effortful strategy.

To address this issue, we tested younger and older adults in a decision-making task where model-based control pays off in terms of higher rewards. We found that older adults showed reduced model-based control compared to younger adults, even in a setting where model-based control leads to better outcomes. Thus, our findings suggest that reduced model-based control in older adults cannot be explained by a reduced willingness to engage in a cognitively effortful strategy when this effort does not pay off. Instead, the effect seems to reflect aging-related limitations in the cognitive processes involved in model-based control.

Previous studies did not investigate how aging affects metacontrol, that is the dynamic adaptation of decision-making strategies to varying situational demands. Therefore, we manipulated the magnitude of rewards and the stability of the task transition structure throughout the task. In line with previous findings (*Kool et al., 2017b*; *Patzelt et al., 2019*), we found that younger adults showed increased model-based control when rewards were amplified, which suggests that they are more willing to incur the cost of planning when incentives are high. We also found that younger adults showed decreased model-based control when the task transition structure was variable, suggesting that repeated changes in the task transition structure increase the demands for model-based control, thus leading to a reduction of this more effortful strategy. We found that older adults' metacontrol was less sensitive to these changes in costs and benefits. They showed less adaptation of model-based control to increased reward magnitudes and to increased demands on structure learning or working memory. Crucially, we also observed reduced metacontrol in a subsample of younger and older adults who were matched for overall reward obtained. This indicates that the reduced adaptation of decision-making strategies is not solely due to older adults having more difficulties with performing the task. These findings point to specific aging-related deficits in the metacontrol of decision-making strategies.

When interpreting our findings, it is important to consider the cross-sectional design of our study. Future research should use longitudinal study designs to better isolate aging-related effects from other cohort-related factors.

## Model-based control and task structure representations in older adults

The reduction of model-based control in older adults may reflect aging-related difficulties in representing the structure of the task. This notion is supported by two further findings: First, older adults showed reduced transition learning rates compared to younger adults. In the reinforcement learning model, the transition learning rate indicates how quickly the mental representation of the task is updated when a change in the task transition structure is observed (e.g., when choosing the blue spaceship now leads to the purple plant while previously it led to the red planet). Since in our task, a new transition reliably signaled a prolonged change in the task transition structure, an optimal decision-maker should immediately update the mental representation of the task accordingly. Lower transition learning rates in older adults indicate that they incorporated information about changes in the task transition structure more slowly (see Materials and methods section for an alternative interpretation), which suggests a more blurred representation of the task structure in older adults.

Second, older adults showed less slowing of second-stage responses in revaluation trials. That is, in trials with a surprising transition due to a change in the task transition structure, older adults showed less slowing of the responses immediately following the surprising transition. Slowed reaction times after surprising transitions have already been reported by other studies using sequential decision-making tasks (*Decker et al., 2016*; *Deserno et al., 2015*; *Shahar et al., 2019*) and could be explained by inhibition of motor and cognitive processes after surprising events (*Wessel and Aron, 2017*). Therefore, the less differentiated responses after revaluation trials and non-revaluation trials in older adults suggest that they had a less accurate representation of the task. Consistent with this line of reasoning, we found that older adults who showed more slowing on the revaluation trials, also showed more model-based control.

There are other explanations for slowed reaction times in response to surprising transitions. For example, they could indicate decreased response vigor due to lower reward expectations (*Deserno et al., 2015*) or higher decision conflict due to less discriminable rewards (*Shahar et al., 2019*). However, we found that the effect of slowed reaction times after surprising transitions

remained even after controlling for model-derived reward expectations. This speaks against the idea that this effect can be explained by reduced response vigor alone. Furthermore, in contrast to the decision-making task used in previous studies, we found slowed reaction times even when there is only one available option at the time of responding (i.e., at the second stage of a trial). Decision conflict would require different choice options and hence cannot account for slowed responses in our study. Therefore, in line with a previous study (*Decker et al., 2016*), we interpret slowed reaction times after surprising transitions as reflecting knowledge of the task structure which was less accurate in older adults.

What are the potential neurobiological foundations of reduced model-based control in older adults? It has been suggested that aging-related alterations in neuromodulation reduce the signal-to-noise ratio in neural information processing and thus reduce representational distinctiveness (*Li et al., 2001*; *Li and Rieckmann, 2014*). This would make it more difficult to maintain and use a representation of the task structure which would impair model-based decision making. Furthermore, prefrontal and hippocampal brain regions are important for the representation of task structures (*Koechlin, 2016*; *Schuck et al., 2016*; *Vikbladh et al., 2019*). Older adults show structural decline of these brain regions (*Raz et al., 2005*; *Resnick et al., 2003*), which might limit their ability to use a task model for model-based decision making. Indeed, underrecruitment of prefrontal brain regions in older adults is linked to impaired learning of task contingencies (*Eppinger et al., 2015*). Moreover, the importance of prefrontal brain regions for the representation of task structure is also in line with findings that transient inhibition of prefrontal areas leads to a reduction in model-based control and in the learning of sequential relations during value-based decision making (*Smittenaar et al., 2013*; *Wittkuhn et al., 2018*). Future research should attempt to link aging-related neurobiological changes more directly to differences in model-based decision making.

## Reduced metacontrol in older adults

In addition to overall reductions in model-based control, we also found that older adults showed reduced adaptation of decision-making strategies to varying reward magnitudes and structure learning or working memory demands. One interpretation of these results could be in terms of aging-related decline in dopaminergic neuromodulation (*Bäckman et al., 2006*) which has been suggested to be involved in the regulation of effortful cognitive processes (*Cools, 2016*; *Westbrook and Braver, 2016*). Alternatively, reduced metacontrol might be related to aging-related tissue loss in brain regions such as dorsal anterior cingulate cortex (*Resnick et al., 2003*) that have been implicated in the modulation of cognitive effort (*Shenhav et al., 2013*; *Silvetti et al., 2018*).

One surprising finding is that older adults showed less adaptation of model-based decision making to differences in the stability of the task transition structure. Why should – in contrast to the effects in younger adults – additional demands to maintain an accurate representation of the task transition structure not further reduce older adults' reliance on model-based decision making? It is conceivable that the comparatively blurred representation of the task structure in older adults in the variable-transitions condition (reflected by lower transition learning rates) could facilitate the use of a (in return, constricted) model-based strategy. In fact, when assuming perfect learning of changes in the task transition structure, we find that even older adults show differences in model-based weights as a function of the transition stability condition (compare to Table S3 in *Supplementary file 1* and *Figure 2—figure supplement 1*). This indicates that older adults show less signs of a model-based strategy with perfect transition learning in variable-transitions blocks than in stable-transitions blocks. This analysis might suggest that the increased demands imposed by repeated changes in the task transition structure could affect other cognitive mechanism that in turn modulate how effortful model-based decision making is.

Metacontrol of decision-making strategies seems to be subject to considerable individual differences. Some participants and especially some older adults showed the opposite of the expected pattern, that is reduced model-based weights in high-stakes compared to low-stakes trials. One possible explanation for this observation is that these participants responded to amplified reward magnitudes with an increased reliance on model-free reward expectations, potentially trying to compensate deficits in increasing reliance on model-based decision-making strategies. Since model-based weights reflect the relative influence of decision-making strategies, increased reliance on model-free strategies would – everything else being equal – result in lower model-based weights.

Our results suggest that older adults are relatively insensitive to costs and benefits when arbitrating between decision-making strategies. However, a recent study by *Yee et al. (2019)* found that incentives boosted performance of both younger and older adults in a cognitive control paradigm, but that these effects are weaker and more gradual in older adults. To understand this difference, note that in our task, the stakes condition was randomly picked on each trial whereas *Yee et al. (2019)* varied incentives over longer blocks of trials. It is possible that older adults take longer to implement effective metacontrol and only show adaptation when task conditions remain stable over a longer period of time.

It is also possible that the older adults were aiming to optimize accuracy rather than reward or a net value of reward and cognitive costs. This would be consistent with the finding that older adults tend to maximize accuracy instead of trading off accuracy against response speed in a perceptual decision making task (*Starns and Ratcliff, 2010*). If older adults were primarily concerned with making an accurate choice, we would not expect different reward magnitudes to affect their reliance on model-based decision making. To investigate this explanation, a promising avenue for research might be to use neurophysiological measurements such as electroencephalography or pupillometry in order to assess the behavioral relevance of stake cues and outcome information to younger and older adults.

It could be argued that the reduced adaptation of model-based decision making to different reward magnitudes in older adults might reflect a general insensitivity to differences in reward in older adults rather than age differences in the metacontrol of decision-making strategies. We think that this alternative explanation is unlikely for two reasons: First, we found that the older adults showed modulation of other model parameters (representing differences in explorative behavior and choice perseveration) in response to the stakes condition, which suggests that they were not generally insensitive to amplified reward magnitudes. Second, we found that the same sample of older adults showed increased performance in response to amplified rewards in an independent cognitive control task. Thus, the age effects seem to be task- and process-specific and do not reflect a general insensitivity to reward. We describe these analyses in more detail in the Appendix 1.

Our study does not address the mechanisms that drive the reduced amounts of metacontrol in older adults. Adapting controlled cognitive processes involves several component functions, such as monitoring information that signals the need to adapt cognitive processes, specification of which cognitive process should be executed and with what intensity, as well as the corresponding regulation of cognitive processes (*Shenhav et al., 2013*). Further studies are required to specify how these processes are affected by aging and contribute to weakened metacontrol.

To conclude, the present study investigated how aging affects model-based decision making and the metacontrol of decision-making strategies according to their costs and benefits. Older adults showed reduced model-based control even when it was associated with increased reward. Thus, our results suggest that aging-related reductions in model-based control reflect cognitive limitations rather than differences in motivation. Moreover, when compared to younger adults, older adults displayed reduced adaptation of decision-making strategies to differences in reward magnitudes and task demands. This cannot be explained in terms of decreased overall task performance, suggesting that the elderly may have specific deficits in the metacontrol of decision-making strategies.

## Materials and methods

### Participants

We tested 63 younger adults and 83 older adults for this study. We excluded data from 22 participants from data analysis: Data from one younger adult and one older adult were excluded because they were unable to respond on more than 50% of the trials in the decision-making task. Data from two older adults were excluded because they did not complete the decision-making task. Data from one older adult were excluded because of consistent perseverative choice behavior on the first stage of the task. Data from 17 older adults were excluded because of sub-threshold performance in a dementia screening test (Montreal Cognitive Assessment with a threshold of 26 points; *Nasreddine et al., 2005*). Thus, the effective sample included 62 younger adults (39 female, age range: 18–30 years, mean age: 22.7 years) and 62 older adults (32 female, age range: 57–80 years,

mean age: 70.2 years). The size of the effective sample was geared to a previous aging study with a similar decision-making task (*Eppinger et al., 2013*).

We assessed participants on different cognitive and motivational variables. Working memory was assessed with the digit span test (forward and backward) (*Wechsler, 1997*), verbal intelligence was assessed with the Spot-A-Word test (*Lindenberger et al., 1993*), processing speed was assessed with the Identical Pictures test (*Lindenberger et al., 1993*) and need for cognition was assessed with the German Need for Cognition short-scale (*Bless et al., 1994*). Consistent with previous findings in larger population-based samples (*Li et al., 2004*), we found reduced fluid abilities (working memory, processing speed) but increased crystallized abilities (verbal intelligence) in older adults. Furthermore, we found no differences between age groups in the self-reported tendency to engage in and enjoy effortful cognitive activities (Need for Cognition). See *Table 1* for details on the psychometric assessment.

All participants were compensated with a baseline payment of 5 € per hour and an additional performance-related payment between 6.83 € and 9.41 € (10 cents for every 60 points obtained in the decision-making task). All participants gave informed written consent. The ethics committee of Technische Universität Dresden approved the study.

## Sequential decision-making task

Every trial started with one of two different first-stage states that were identified by a pair of spaceships (see *Figure 1*). Spaceships were displayed side by side on the screen and each spaceship appeared on the left or the right side equally often. Throughout the experiment, pairings of spaceships remained constant. After selecting one spaceship, the participant transitioned to one of two second-stage states that were represented by two different planets (a red planet and a purple planet) and a corresponding alien. Both second-stage states were associated with a scalar reward ('space treasure') that the participant received after selecting the alien. Similar to previous studies, the reward magnitude available in each second-stage state drifted over the course of the experiment (Gaussian random walks with mean $\mu = 0$, standard deviation $\sigma = 2$ and reflecting boundaries at 0 and 9; reward values were rounded to integers).

The transition to the second-stage state (planet) was contingent on the choice of the spaceship on the first stage. In both first-stage states, the choice of one spaceship led to the red planet and the choice of the other spaceship led to the purple planet (i.e. the transition was deterministic). In stable-transitions blocks, the spaceships' destinations stayed the same over the course of the entire block. In contrast, during variable-transitions blocks, every 6 to 14 trials the spaceships in one of the two starting states swapped their destination. While participants were informed at the beginning of each trial block whether transitions were stable or variable, they were not informed when these changes would happen.

At the beginning of each trial, a stake cue informed the participant about the stakes condition of the current trial. The stakes condition determined how reward was converted into points. In low-

**Table 1.** *Psychometric assessment of complete sample.*

| Variable | N assessed (younger/older adults) | Mean (SD) younger adults | Mean (SD) older adults | Bayes Factor[*] |
|---|---|---|---|---|
| Working memory | 62/61 | 18.42 (3.80) | 15.95 (4.22) | 31.88 |
| Verbal intelligence | 60/60 | 69.07% (11.82) | 81.94% (5.71) | >100 |
| Processing speed (reaction times) | 61/60 | 2032.17 s (315.87) | 3319.93 s (516.42) | >100 |
| Processing speed (accuracy) | 61/60 | 94.10% (6.02) | 94.84% (5.48) | 0.24 |
| Need for cognition | 61/59 | 82.18 (13.25) | 81.44 (13.24) | 0.20 |

[*]Bayes Factors quantify the evidence in favor of one hypothesis (here: non-equal means) as opposed to a competing hypothesis (here: equal means). Commonly, Bayes Factors between 3 and 10 are considered as representing moderate evidence for the hypothesis and Bayes Factor above 10 as representing strong evidence for the hypothesis. Conversely, Bayes Factors between 1/3 and 1/10 represent moderate evidence for the competing hypothesis and Bayes Factors below 1/10 represent strong evidence for the competing hypothesis (*Lee and Wagenmakers, 2013*). We calculated Bayes Factors with the BayesFactors package (*Morey et al., 2015*).

DOI: https://doi.org/10.7554/eLife.49154.007

stakes trials, the obtained reward was multiplied by a factor of 1. In high-stakes trials, the obtained reward was multiplied by a factor of 5. The total point count was displayed in the top-right corner of the screen during the entire experiment.

The experiment consisted of 320 trials that were divided into four blocks of 80 trials: two stable-transitions blocks and two variable-transitions blocks. Stable-transitions blocks and variable-transitions blocks alternated. In each block, half of the trials were low-stakes trials and the other half of the trials were high-stakes trials, appearing in random order and being counter-balanced over the two first-stage states.

In order to decrease variability in task performance between subjects, we created four trial sequences that determined the sequence of first-stage states, second-stage rewards, stakes conditions, and transition variability conditions. Each participant performed the task with one of these four trials sequences and trial sequences were counter-balanced across age groups. Across all trial sequences, the first-stage states and the reward trajectories for the two second-stage states were identical. Stakes condition and transition variability condition were counter-balanced across trial sequences.

The experiment was implemented in Matlab and run on a standard PC. Participants controlled the experiment with a computer keyboard. The left and the right spaceships were selected with the keys F and J, respectively, and the second-stage response was made using the space bar. If participants missed the response deadline of 3 s (first-stage state) or 2 s (second-stage state), the trial was aborted, no reward was obtained and the experiment proceeded with the next trial. Trials with missed responses were excluded from the analysis. Trials with missing second-stage responses were included in the parameter estimation of the reinforcement-learning model because they are potentially informative with respect to the changes in the transition structure. For younger adults, 0.2% of trials were aborted at the first stage and 1.2% were aborted at the second stage. For older adults, 1.6% of trials were aborted at the first stage and 2.1% were aborted at the second stage. If responses happened in time, the selected option was highlighted for the remaining time of the response window such that trial duration remained constant for all participants.

Prior to the experiment, participants were instructed extensively about the reward distribution, the transition structure and the stakes manipulation. To ensure that participants understood the task structure, we asked participants to pick the correct spaceship leading to a given planet up to a performance criterion of 10 correct consecutive choices of the two planets. Furthermore, participants had to report the correct number of earned points (combination of reward magnitude and stakes cue) on 10 consecutive correct answers in a row. During this instruction procedure, they also completed 20 practice trials with stable transitions and 20 practice trials with variable transitions.

## Computational model

We used an established hybrid reinforcement-learning model (*Daw et al., 2011*; *Gläscher et al., 2010*; *Kool et al., 2017b*) to describe participants' behavior in the decision-making task. In the task, every trial $t$ started in one of two first-stage states ($s_{1,t}$) where one of two possible actions ($a_{1,t}$) could be selected. Depending on the action, the participant deterministically transitioned to a second-stage state ($s_{2,t}$) where only one action ($a_{2,t}$) was available and a reward ($r_t$) obtained. The model consisted of a model-free and a model-based learner that both learned an expectation of long-term future reward $Q(s,a)$ for each combination of state and action.

*Model-free learner.* We assumed that the model-free learner assigned an intermediate reward expectation of 4.5 (as the arithmetic mean of the minimum and the maximum possible reward) to all actions in all states at the beginning of the experiment. The model-free learner then used the SARSA($\lambda$) temporal difference learning algorithm (*Rummery and Niranjan, 1994*) to update reward expectations based on the discrepancy between the expected and experienced reward. In our task, this meant that at every stage, a reward prediction error $\delta$ was computed according to

$$\delta_{1,t} = Q_{MF}(s_{2,t}, a_{2,t}) - Q_{MF}(s_{1,t}, a_{1,t})$$

$$\delta_{2,t} = r_t - Q_{MF}(s_{2,t}, a_{2,t})$$

This reward prediction error was then used to update reward expectations at the first and second stage:

$$Q_{\mathrm{MF}}\big(s_{1,t},a_{1,t}\big) \leftarrow\ Q_{\mathrm{MF}}\big(s_{1,t},a_{1,t}\big) + \alpha\delta_{1,t} + \alpha\lambda\delta_{2,t}$$

$$Q_{\mathrm{MF}}\big(s_{2,t},a_{2,t}\big) \leftarrow\ Q_{\mathrm{MF}}\big(s_{2,t},a_{2,t}\big) + \alpha\delta_{2,t}$$

Here, $\alpha$ is the reward learning rate (ranging between 0 and 1) that determined how strongly new information was incorporated into the reward expectations. $\lambda$ is the eligibility trace decay (ranging between 0 and 1) that determined to what extent a reward prediction error affected the update of reward expectations for past actions (here, how the reward prediction error at the second stage affected the update of the reward expectations at the first stage).

*Model-based learner.* The model-based learner predicted reward by learning a transition probability $T(s_2|s_1,a_1)$ that represented the probability of transitioning to the second-stage state $s_2$ after selecting action $a_1$ in the first-stage state $s_1$. The reward expectations at the second-stage were then weighted by the transition probabilities:

$$Q_{MB}\big(s_{1,t},a_{1,t}\big) = \sum_{s_2} T\big(s_2|s_{1,t},a_{1,t}\big) Q_{MB}(s_2,a_2)$$

Model-based reward expectations at the second stage are identical to second-stage model-free reward expectations because they are both an estimate of the immediate reward. Therefore, we set

$$Q_{\mathrm{MB}}\big(s_{2,t},a_{2,t}\big) =\ Q_{\mathrm{MF}}\big(s_{2,t},a_{2,t}\big)$$

Given that all participants practiced the task with variable transitions, we assumed that participants started without knowledge of the correct initial transition structure, such that $T(s_2|s_1,a_1)=0.5$ for all values of $s_1$, $a_1$, and $s_2$. After transitioning to a second-stage state, a state prediction error $\delta^{SPE}$ was computed as

$$\delta_t^{SPE} = 1 - T\big(s_{2,t}|s_{1,t},a_{1,t}\big)$$

This state prediction error was used to update the transition probability for the experienced transition. Moreover, the transition probability to the alternative second-stage state ($\neg s_{2,t}$) had to be reduced to ensure that the sum of the both transition probabilities still equaled 1:

$$T\big(s_{2,t}|s_{1,t},a_{1,t}\big) \leftarrow\ T\big(s_{2,t}|s_{1,t},a_{1,t}\big) + \eta\delta_t^{SPE}$$

$$T\big(\neg s_{2,t}|s_{1,t},a_{1,t}\big) \leftarrow\ T\big(\neg s_{2,t}|s_{1,t},a_{1,t}\big)(1-\eta)$$

Here, $\eta$ is the transition learning rate (ranging between 0 and 1) that defined how quickly transition probabilities were updated. In stable-transitions blocks, we set $\eta = 1$ which is consistent with previous implementations for this task where no changes in the transition structure occurred; thus $\eta$ was a free parameter only in variable-transitions blocks.

Due to the anti-correlated transition structure of the task (the two actions in a first-stage state led always to different second-stage states), it was possible to make an inference where the alternative action in the first-stage state $\neg a_{1,t}$ would have led to. We therefore implemented a counterfactual update of the transition probabilities, similar to the update based on the experienced transition described above:

$$\delta_t^{CF-SPE} = 1 - T\big(s_{2,t}|s_{1,t},\neg a_{1,t}\big)$$

$$T\big(\neg s_{2,t}|s_{1,t},\neg a_{1,t}\big) \leftarrow\ T\big(\neg s_{2,t}|s_{1,t},\neg a_{1,t}\big) + \eta_{CF}\delta_t^{CF-SPE}$$

$$T\big(s_{2,t}|s_{1,t},\neg a_{1,t}\big) \leftarrow\ T\big(s_{2,t}|s_{1,t},\neg a_{1,t}\big)(1-\eta_{CF})$$

Here, $\eta_{CF}$ is the transition learning rate for counterfactual transitions. We set $\eta_{CF} = \eta$ because model comparisons indicated higher goodness of fit for a model with a single transition learning rate based on the Akaike information criterion in both age groups (see Table S4 in *Supplementary file 1*).

While this model assumes incremental learning of changes in the task transition structure, it is also conceivable that an agent maintains representations of the four possible task transition structures in working memory and activates them according to the observed transitions. With the current task design, we cannot dissociate between these two mechanisms. In both cases, higher transition learning rates reflect better abilities to use an accurate representation of the task transition structure because any inaccuracies in the use of a task model would be forced upon this model parameter.

*Choice rule.* Model-free and model-based reward expectations in the first stage were integrated using a model-based weight $\omega$ (ranging from 0 to 1):

$$Q_{net}(s_1, a_1) = (1 - \omega)Q_{MF}(s_1, a_1) + \omega Q_{MB}(s_1, a_1)$$

We used a softmax function to map reward expectations to choice probabilities:

$$P(a_{1,t} = a_1 | s_{1,t}) = \frac{\exp(\beta[Q_{net}(s_{1,t}, a_1) + \pi \cdot rep(a_1) + \rho \cdot resp(a_1)])}{\sum_{a'_1} \exp(\beta[Q_{net}(s_{1,t}, a'_1) + \pi \cdot rep(a'_1) + \rho \cdot resp(a'_1)])}$$

Here, $\beta$ is the inverse softmax temperature (left-bounded to 0) that controlled how strongly choice probabilities were guided by reward expectations. The variable $rep(a_1)$ was defined as one if $a_1$ was the action that was chosen in the previous trial, and 0 otherwise. The choice stickiness parameter $\pi$ (unbounded) captured choice perseveration ($\pi > 0$) or choice switching ($\pi < 0$). The variable $resp(a_1)$ was defined as one if the action $a_1$ could be selected with the same response key as the response key that was used in the previous trial, and 0 otherwise. The response stickiness parameter $\rho$ captured perseveration ($\rho > 0$) or switching ($\rho < 0$) of the response key at the first stage.

*Model fitting procedure.* For each participant, we obtained maximum a posteriori estimates of the free model parameters. For parameter estimation, we used the mfit toolbox in Matlab (*Gershman, 2016*). For all parameters bounded between 0 and 1 ($\alpha$, $\lambda$, $\eta$, $\omega$), we used a Beta(2,2) prior, for the inverse softmax temperature $\beta$, we used a Gamma(3,0.2) prior, and for the two stickiness parameters $\pi$ and $\rho$, we used Normal(0,1) priors. To avoid local optima, we ran the optimization procedure 100 times for each participant with random initializations of the parameters and selected the parameters of the run with the highest posterior probability.

To detect differences in decision-making strategies between task conditions, a different model-based weight was estimated for each combination of stakes condition and transition condition. Accordingly, the model contained ten free parameters: $\alpha$, $\lambda$, $\eta$, $\omega_{low,stable}$, $\omega_{high,stable}$, $\omega_{low,variable}$, $\omega_{high,variable}$, $\beta$, $\pi$, and $\rho$. Analyses on parameter recovery (see Appendix 2) showed that the identifiability of model parameters with our task design was substantial and similar to previous versions of the sequential decision-making task (cf., *Kool et al., 2016*). A posterior predictive check (see Appendix 2) revealed a good absolute (as opposed to relative) model fit because the model could reproduce individual differences in observed behavior.

In the exhaustive version of the model, we varied all parameters between all four task conditions (apart from η which was only varied between low-stakes trials and high-stakes trials in the variable-transitions blocks, see above), leading to overall 26 free parameters.

## Hierarchical regression analysis

Hierarchical Bayesian regression analyses were conducted in R Studio using the brms package (*Bürkner, 2017*). This package interfaces with the probabilistic programming language Stan and draws samples from the posterior distribution over model parameters by means of a Markov Chain Monte Carlo procedure and a NUTS sampling algorithm. For each model, we ran four independent chains with 2000 iterations each, of which the first 1000 were discarded as warm-up, leaving 4000 posterior samples in total. If effective sample size was below 400 for reported parameters, we increased the number of iterations accordingly. We assessed the convergence of all chains via the Gelman-Rubin statistic (R-hat <1.1) for all parameters.

All regression models included a random-by-participant intercept and, where multiple data points for each participant and condition were available (i.e., in the analysis of reaction times), we additionally included a random-by-participant slope for all within-subject factors and their interaction terms. When there was an equal number of data points per condition, we used effect coding for categorical variables (younger adults = 0.5, older adults = −0.5, low-stakes = −0.5, high-stakes = 0.5, stable-transitions = 0.5, variable-transitions = −0.5) which allows interpreting regression coefficients as

main and interaction effects. In the analysis of revaluation trials, categorical variables were dummy-coded (younger adults = 0, older adults = 1, non-revaluation trial = 0, revaluation trial = 1) due to the unequal number of revaluation trials and non-revaluation trials. Moreover, in the analysis of revaluation trials, we only considered trials from variable-transitions blocks and excluded trials with reaction times below 200 ms when analyzing differences in reaction times. Weakly informative default priors were used.

Each reported regression coefficient is described by the mean of its marginal posterior distribution $\bar{\beta}$ and the 95% credible interval that is computed as the [.025,.975] percentile interval and can be interpreted as including the parameter of interest with 95% probability. A full report on all group-level coefficients can be found in the Tables S5-S21 in *Supplementary file 1*.

## Performance-matching procedure

We matched participants between age groups based on their mean baseline-corrected reward (i.e., the difference between obtained reward and the mean reward available at the two planets, before multiplication in high-stakes trials) by assigning to each participant a candidate partner from the respective other age group such that the absolute difference in mean baseline-corrected rewards between both participants was minimized. Pairs of participants who were mutually assigned as each other's candidate partner were included into the performance-matched sample.

This procedure yielded a subsample of 26 younger adults (14 female, age range: 18–27 years, mean age: 22.4 years) and 26 older adults (14 female, age range: 57–79 years, mean age: 69.1 years).

## Additional information

### Funding

| Funder | Grant reference number | Author |
|---|---|---|
| German Research Foundation | SFB 940/2 B7 | Ben Eppinger |

The funders had no role in study design, data collection and interpretation, or the decision to submit the work for publication.

### Author contributions

Florian Bolenz, Conceptualization, Resources, Data curation, Software, Formal analysis, Validation, Investigation, Visualization, Methodology, Writing—original draft, Writing—review and editing; Wouter Kool, Resources, Software, Methodology, Writing—review and editing; Andrea MF Reiter, Conceptualization, Methodology, Writing—review and editing; Ben Eppinger, Conceptualization, Resources, Software, Supervision, Funding acquisition, Methodology, Project administration, Writing—review and editing

### Author ORCIDs

Florian Bolenz https://orcid.org/0000-0003-2213-1071

### Ethics

Human subjects: All participants gave informed written consent. The ethics committee of Technische Universität Dresden approved the study (reference number EK 519122015).

### Decision letter and Author response

Decision letter https://doi.org/10.7554/eLife.49154.017
Author response https://doi.org/10.7554/eLife.49154.018

## Additional files

### Supplementary files
• Supplementary file 1. Tables for model parameters, model comparison and statistical results.
DOI: https://doi.org/10.7554/eLife.49154.008
• Transparent reporting form
DOI: https://doi.org/10.7554/eLife.49154.009

### Data availability
Experimental data as well as analysis scripts are available online at https://osf.io/xne7c/.

The following dataset was generated:

| Author(s) | Year | Dataset title | Dataset URL | Database and Identifier |
|---|---|---|---|---|
| Bolenz F, Kool W, Reiter AMF, Eppinger B | 2019 | Metacontrol of decision-making strategies in human aging | https://doi.org/10.17605/OSF.IO/XNE7C | Open Science Framework, 10.17605/OSF.IO/XNE7C |

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

# Appendix 1

DOI: https://doi.org/10.7554/eLife.49154.010

## Analysis of further model parameters in the exhaustive reinforcement-learning model

We analyzed differences in model parameters other than model-based weights for older adults in order to assess whether older adults modulate other decision-making processes than the reliance on model-based learning when reward magnitudes and task demands change. For this analysis, we resorted to the parameter estimates from the exhaustive reinforcement-learning model which allowed all parameters to vary between experimental conditions.

For the inverse softmax temperature, we found a main effect of transition stability condition ($\bar{\beta}_{\text{transition}}$ = 0.10, 95% CI = [0.04, 0.16]) indicating more exploitive and less explorative behavior in stable-transitions blocks, as well as moderate evidence for a main effect of stakes condition ($\bar{\beta}_{\text{stakes}}$ = 0.04, 95% CI = [-0.02, 0.10], with 91.3 % of the posterior probability mass above zero) suggesting more exploitive and less explorative behavior in high-stakes trials.

For the choice stickiness parameter, we observed a main effect of transition stability condition ($\bar{\beta}_{\text{transition}}$ = -0.21, 95% CI = [-0.34, -0.07]) indicating higher choice perseveration in variable-transitions blocks as well as moderate evidence for a main effect of stakes condition ($\bar{\beta}_{\text{stakes}}$ = 0.11, 95% CI = [-0.02, 0.24], with 95.2 % of the posterior probability mass above zero) suggesting higher choice perseveration in high-stakes trials. We did not obtain conclusive evidence for differences between task conditions for other model parameters. These results indicate that older adult might still adapt other cognitive processes than model-based decision making to varying task demands and reward magnitudes.

## Analysis of reward effects in a cognitive control task

To investigate whether reduced metacontrol of decision-making strategies might be accounted for by a general aging-related insensitivity to differences in rewards, we analyzed behavior in a task-switching task with varying rewards (*Otto and Daw, 2019*). This task was assessed as part of a parallel study in the same sample. Results of this task will be reported elsewhere (Devine et al., in preparation). One older adult who participated in the sequential decision-making task was not assessed on the task-switching task, resulting in an effective sample of 62 younger adults and 61 older adults for this experiment.

Participants were instructed to perform two tasks: the 'food task' and the 'size task' (see *Appendix 1—figure 1*). In the food task, participants had to judge whether the stimulus on-screen was a fruit or a vegetable. In the size task, they had to judge whether the stimulus on-screen was small or large. Stimuli consisted of pictures of fruits (apple, pear) and vegetables (cucumber, eggplant) where each stimulus could appear in a large-scale version or a small-scale version. Each trial started with a fixation cross (200 ms) followed by a reward cue (between 850 and 1350 ms). The reward cue signaled the number of points that could be earned by a correct response in the current trial. Points varied from trial to trial according to Gaussian random walks with standard deviation 30 and reflecting boundaries at 5 and 95. The reward cue was followed by a second fixation cross (200 ms). Then, a task cue was presented (500 ms) indicating to which dimension of the stimulus participants needed to respond (category or size). Trials in which participants needed to respond to the same dimension as in the preceding trial were defined as *stay trials*, and trials in which participants needed to respond to the other dimension than in the preceding trial were defined as *switch trials*. Next, one stimulus was presented until a response was given (using the A and L key on a standard computer keyboard) or up to max. 750 ms for younger adults and max. 1125 ms for older adults. A trial ended with a response feedback presented for 1000 ms.

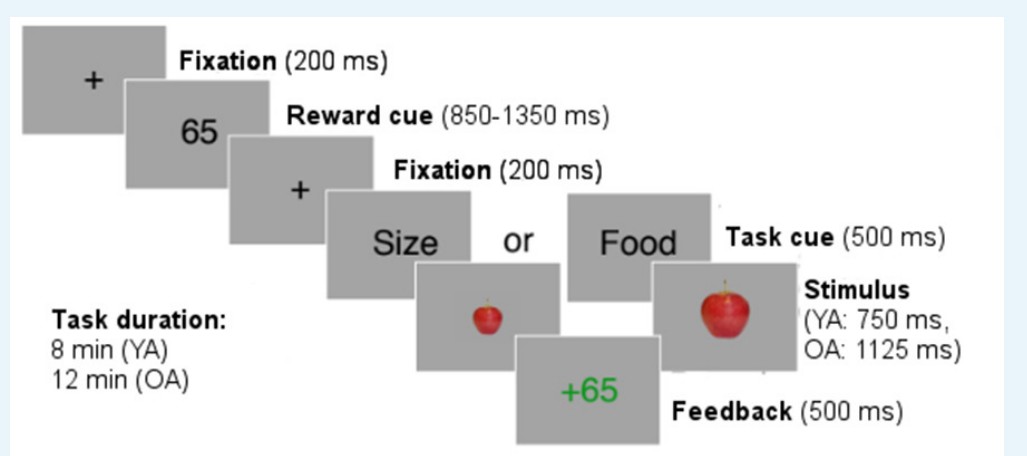

**Appendix 1—figure 1.** Structure of the task-switching task.
DOI: https://doi.org/10.7554/eLife.49154.011

The task lasted for a fixed amount of time (8 min for younger adults, 12 min for older adults) and we recorded 8412 trials in younger adults and 11636 trials in older adults. Timeout trials were excluded from the analysis (3.51% in younger adults, 6.38% in older adults). Participants received monetary compensation for points earned in the task (3 cents for 100 points).

We analyzed the effect of age group and points at stake on reaction times using a hierarchical regression analysis. To analyze differences in the effect of points at stake, we also included an interaction term of age group and points at stake. Consistent with previous work (*Otto and Daw, 2019*), we controlled for trial type (stay/switch), task (category/size), trial type of the previous trial, error in the previous trial, same response as in the previous trial, trial number, and duration of the reward cue. Trials with reaction times below 200 ms were excluded from the analysis. Reaction times were log-transformed and all continuous predictor variables were z-standardized.

We found a negative regression weight for points at stake for older adults ($\bar{\beta}_{\text{points(older adults)}}=-0.005$, 95% CI = $[-0.009,-0.001]$) indicating that older adults responded faster when more points were at stake. We did not obtain evidence for an effect of points at stake for younger adults ($\bar{\beta}_{\text{points(younger adults)}}=0.000$, 95% CI = $[-0.004, 0.005]$).

These analyses indicate that older people adapt their reaction times in a cognitive control task to differences in reward magnitude. This speaks against the idea that older adults are generally insensitive to differences in reward magnitude and is consistent with other behavioral and neuroimaging findings (e.g., *Samanez-Larkin et al., 2007*).

## Appendix 2

DOI: https://doi.org/10.7554/eLife.49154.010

### Analysis of parameter recovery

In order to assess whether the reinforcement-learning model is capable of reliably identifying the reliance on model-free and model-based decision making in our version of the sequential decision-making task, we used the generative version of the model to simulate the behavior of 500 agents. For each agent, we sampled model parameters (*true parameters*) randomly from uniform distributions: $\{\alpha, \lambda, \eta, \omega_{low,stable}, \omega_{high,stable}, \omega_{low,variable}, \omega_{high,variable}\} \sim U(0,1)$, $\beta \sim U(0,2)$, $\{\pi, \rho\} \sim U(-0.5, 0.5)$ (cf., **Kool et al., 2016**). Next, we used our model-fitting procedure (as described in the Materials and methods section) to obtain maximum a posteriori estimates of model parameters for each simulated agent from its choice behavior (*estimated parameters*).

We found substantial correlations between true and estimated parameters for model-based weights in all conditions (low-stable: $r = 0.76$, high-stable: $r = 0.75$, low-variable: $r = 0.79$, high-variable: $r = 0.76$). This indicates that our method can extract meaningful parameter estimates for the model-based weights. For the other model parameters, we found the following correlations: reward learning rate $\alpha$, $r = 0.89$, eligibility trace decay $\lambda$, $r = 0.76$, transition learning rate $\eta$, $r = 0.74$, inverse softmax temperature $\beta$, $r = 0.92$, choice stickiness $\pi$, $r = 0.67$, response stickiness $\rho$, $r = 0.71$.

### Analysis of absolute model fit

We performed a posterior predictive check in order to validate whether the reinforcement-learning model was able to reproduce behavioral patterns we observed in the empirical data. For each participant, we simulated 50 data sets using the individual parameter estimates and calculated the mean baseline-corrected reward across these data sets. *Appendix 2—figure 1* shows that the simulated data reproduced the empirically observed individual differences in both age groups. The correlation between empirical and simulated values was $r = 0.90$ across the complete sample, $r = 0.89$ for the subsample of younger adults and $r = 0.87$ for the subsample of older adults.

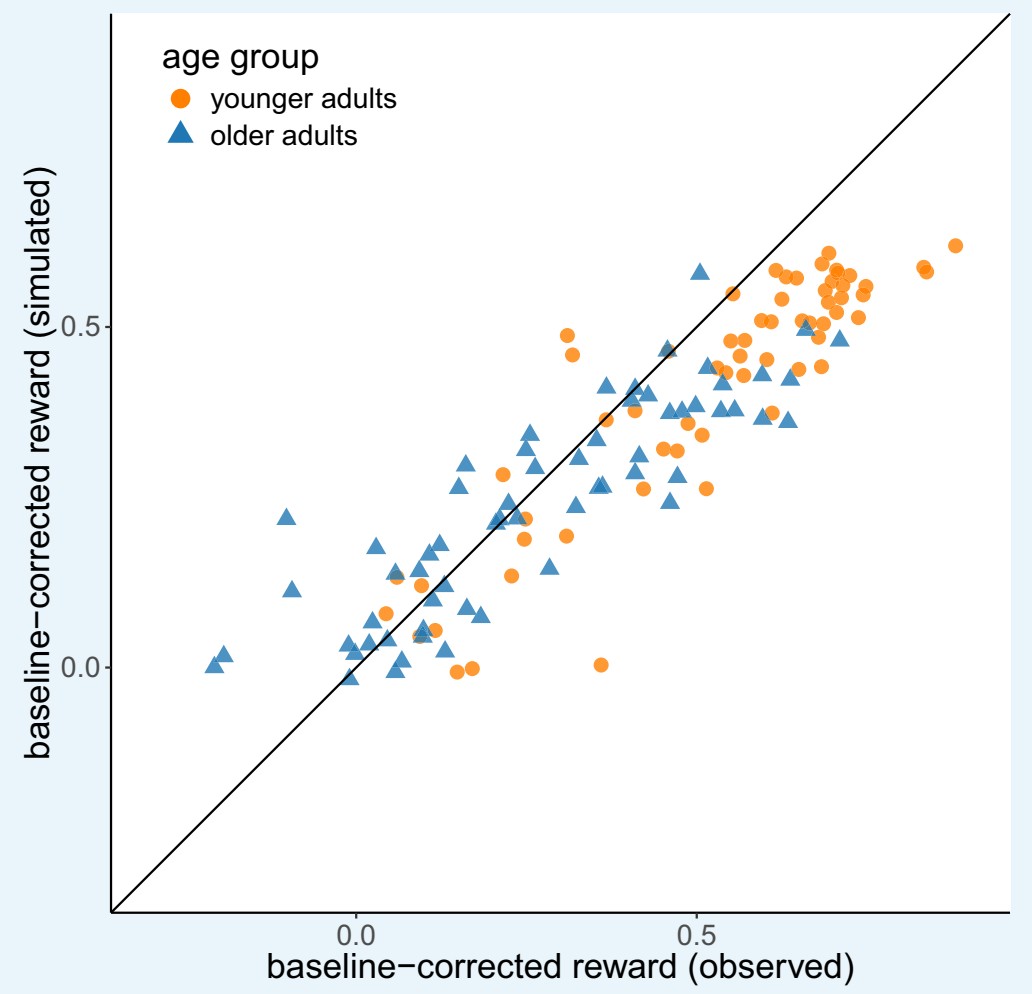

**Appendix 2—figure 1.** Posterior predictive check. Comparison of simulated vs. empirically observed values for baseline-corrected reward. The diagonal represents the identity line.
DOI: https://doi.org/10.7554/eLife.49154.013

