## [Decision Letter]

Thank you for submitting your article "Metacontrol of decision-making strategies in human aging" for consideration by *eLife*. Your article has been reviewed by three peer reviewers, one of whom is a member of our Board of Reviewing Editors, and the evaluation has been overseen by a Reviewing Editor and Christian Büchel as the Senior Editor. The following individuals involved in review of your submission have agreed to reveal their identity: Greg Samanez-Larkin (Reviewer #2).

The reviewers have discussed the reviews with one another and the Reviewing Editor has drafted this decision to help you prepare a revised submission.

Summary:

This manuscript presents new data and results comparing younger and older adults in a reinforcement learning task to provide (1) additional evidence that older adults are less model-based than younger adults consistent with prior research and (2) new evidence that older adults are less likely to adjust the use of model-based decision making even when doing so would improve overall task performance. The new analyses of potential differences in reward sensitivity, the parameter recovery analysis, and the performance-matched comparisons strengthen the paper and provide additional evidence for the conclusion of age differences in model-based behavior.

Essential revisions:

- Subsection “Model-based control and task structure representations in older adults”: Why is it "'implicit' knowledge of the task structure"? Could it not also be 'explicit' knowledge? Reviewer #1 thinks it would be best to just say 'knowledge' and leave it open whether it is implicit or explicit.

- Reviewer #1 thinks it would be important to clearly state that the current cross-sectional study does not isolate effects of age from those of cohort. The cross-sectional design should be mentioned in the abstract.

- Reviewer #2 wonders if the authors make anything of the quite strong individual differences within older adults. Figure 2 shows relatively flat mean lines but that one third to half of the older adults show patterns similar to the young averages. Yet there are also quite a few – a handful on the left and many on the right – that show a strong opposite pattern compared to the younger adults. The mean lines suggest that older adults on average show no differences when really there are subgroups showing opposite effects. Reviewer #2 thinks it could be useful to ads some mention of it – assuming the authors are confident in the accuracy of the parameter estimates. Why would some individuals show these strong opposite effects where they become much less model based when the stakes are higher?

- Figure 2 should have an overall performance plot like Figure 3A. It's very helpful to see the distributions and mean differences in overall performance plotted.

- In several places within the manuscript, the authors suggest that task structure representations may grow "less distinct" with aging. Reviewer #3 has troubles understanding what is meant by this phrase. Distinct from what?

- The model of learning in the variable transitions condition assumes that transition structure knowledge is updated via an incremental process governed by a learning rate. Given that transitions in the task are fully deterministic, and thus observing a single change provides full information, it seems possible that the task structure representation in the variable transition condition might rely more on a working memory process, in which the four transition mappings are actively maintained and updated when revaluation trials are observed. Perhaps with older adults having fewer "slots" or a faster decay rate. Reviewer #3 is not sure that these two models would be formally discriminable/identifiable, but the independent working memory measure collected might be informative. Does WM correlate with RT slowing on revaluation trials and overall performance in the variable transitions condition? This sort of analysis might provide insight into whether that the aging-related cognitive changes that impair task performance in older adults are more centrally related to structure learning, versus working memory.

- The absence of a difference in performance in older adults between the stable and variable transitions condition merits further discussion. The discussion at present highlights differences between young and older adults' performance in the variable transition condition, suggesting that the relative performance at different ages suggests age-related impairments in structure learning. But it is also important to consider why the variable transition condition does not appear to impose a cost on older adults' model-based evaluation, as hypothesized. What are the cognitive limitations on older adults' performance in the stable condition, when the structure learning demands of the task are relatively simple?

- Descriptive statistics for the parameter estimates for younger and older adults should be reported in a table. At present, the age group effects are reported, but not the estimates themselves.

---

## [Author Response]

Summary:This manuscript presents new data and results comparing younger and older adults in a reinforcement learning task to provide (1) additional evidence that older adults are less model-based than younger adults consistent with prior research and (2) new evidence that older adults are less likely to adjust the use of model-based decision making even when doing so would improve overall task performance. The new analyses of potential differences in reward sensitivity, the parameter recovery analysis, and the performance-matched comparisons strengthen the paper and provide additional evidence for the conclusion of age differences in model-based behavior.Essential revisions:- Subsection “Model-based control and task structure representations in older adults”: Why is it "'implicit' knowledge of the task structure"? Could it not also be 'explicit' knowledge? Reviewer #1 thinks it would be best to just say 'knowledge' and leave it open whether it is implicit or explicit.

We agree and we have changed this sentence accordingly.

- Reviewer #1 thinks it would be important to clearly state that the current cross-sectional study does not isolate effects of age from those of cohort. The cross-sectional design should be mentioned in the abstract.

We address this point now clearly in the Abstract and Discussion section:

Abstract:

“In this cross-sectional study, we tested younger and older adults in a sequential decision-making task that dissociates model-free and model-based strategies.”

Discussion section:

“When interpreting our findings, it is important to consider the cross-sectional design of our study. Future research should use longitudinal study designs to better isolate aging-related effects from other cohort-related factors.”

- Reviewer #2 wonders if the authors make anything of the quite strong individual differences within older adults. Figure 2 shows relatively flat mean lines but that one third to half of the older adults show patterns similar to the young averages. Yet there are also quite a few – a handful on the left and many on the right – that show a strong opposite pattern compared to the younger adults. The mean lines suggest that older adults on average show no differences when really there are subgroups showing opposite effects. Reviewer #2 thinks it could be useful to ads some mention of it – assuming the authors are confident in the accuracy of the parameter estimates. Why would some individuals show these strong opposite effects where they become much less model based when the stakes are higher?

We address this point now in the Discussion section:

“Metacontrol of decision-making strategies seems to be subject to considerable individual differences. Some participants and especially some older adults showed the opposite of the expected pattern, that is reduced model-based weights in high-stakes compared to low-stakes trials. One possible explanation for this observation is that these participants responded to amplified reward magnitudes with an increased reliance on model-free reward expectations, potentially trying to compensate deficits in increasing reliance on model-based decision-making strategies. Since model-based weights reflect the relative influence of decision-making strategies, increased reliance on model-free strategies would – everything else being equal – result in lower model-based weights.”

- Figure 2 should have an overall performance plot like Figure 3A. It's very helpful to see the distributions and mean differences in overall performance plotted.

We edited Figure 2 accordingly and added an additional analysis referring to overall task performance (Results section):

“We compared the overall baseline-corrected reward obtained between age groups (Figure 2A). An effect-coded hierarchical regression analysis revealed an age-difference in reward (β_age group_ = 0.25, 95% credible interval (CI) = [0.17, 0.33]), indicating that younger adults performed on average better than older adults.”

- In several places within the manuscript, the authors suggest that task structure representations may grow "less distinct" with aging. Reviewer #3 has troubles understanding what is meant by this phrase. Distinct from what?

To make our intended meaning clearer, we replaced “distinct” with “accurate”.

- The model of learning in the variable transitions condition assumes that transition structure knowledge is updated via an incremental process governed by a learning rate. Given that transitions in the task are fully deterministic, and thus observing a single change provides full information, it seems possible that the task structure representation in the variable transition condition might rely more on a working memory process, in which the four transition mappings are actively maintained and updated when revaluation trials are observed. Perhaps with older adults having fewer "slots" or a faster decay rate. Reviewer #3 is not sure that these two models would be formally discriminable/identifiable, but the independent working memory measure collected might be informative. Does WM correlate with RT slowing on revaluation trials and overall performance in the variable transitions condition? This sort of analysis might provide insight into whether that the aging-related cognitive changes that impair task performance in older adults are more centrally related to structure learning, versus working memory.

The reviewer raises a very interesting question. When analyzing the correlation between working memory and reaction time slowing in revaluation trials, we find that (z-standardized) working memory scores predict reaction time differences between revaluation trials and non-revaluation trials for younger adults (β = 0.03, 95% CI = [0.01, 0.06]) but not for older adults (β = 0.00, 95% CI = [-0.03, 0.03]). Moreover, we find moderate evidence that working memory scores predict task performance in variable-transitions blocks in younger adults (β = 0.05, 95% CI = [-0.00, 0.11]) but not in older adults (β = 0.01, 95% CI = [-0.05, 0.07]). While these results suggest an association between working and the use of an accurate task structure representation in the younger adults, this association is less clear for older adults. Importantly, our results do not require any commitment to one of the two mechanisms (structure learning vs. working memory), in both cases higher transition learning rates would reflect better abilities to uses an accurate representation of the task structure.

We address this point now in the Materials and methods section:

“While this model assumes incremental learning of changes in the task transition structure, it is also conceivable that an agent maintains representations of the four possible task transition structures in working memory and activates them according to the observed transitions. With the current task design, we cannot dissociate between these two mechanisms. In both cases, higher transition learning rates reflect better abilities to use an accurate representation of the task transition structure because any inaccuracies in the use of a task model would be forced upon this model parameter.”

We also point to this alternative interpretation when discussing transition learning rates in the Results section and the Discussion section. Moreover, we mention working memory demands along with structure learning demands when discussing differences between stable-transitions and variable-transitions blocks, e.g.:

“We found that the older adults’ metacontrol was less sensitive to these changes in costs and benefits. They showed less adaptation of model-based control to increased reward magnitudes and to increased demands on structure learning or working memory.”

- The absence of a difference in performance in older adults between the stable and variable transitions condition merits further discussion. The discussion at present highlights differences between young and older adults' performance in the variable transition condition, suggesting that the relative performance at different ages suggests age-related impairments in structure learning. But it is also important to consider why the variable transition condition does not appear to impose a cost on older adults' model-based evaluation, as hypothesized. What are the cognitive limitations on older adults' performance in the stable condition, when the structure learning demands of the task are relatively simple?

We address this point now in the Discussion section and have added an additional analysis to the supplementary information (Supplementary file 1—table S3 and Figure 2 —figure supplement 1).

“One surprising finding is that older adults showed less adaptation of model-based decision making to differences in the stability of the task transition structure. Why should – in contrast to the effects in younger adults – additional demands to maintain an accurate representation of the task transition structure not further reduce older adults’ reliance on model-based decision making? It is conceivable that the comparatively blurred representation of the task structure in older adults in the variable-transitions condition (reflected by lower transition learning rates) could facilitate the use of a (in return, constricted) model-based strategy. In fact, when assuming perfect learning of changes in the task transition structure, we find that even older adults show differences in model-based weights as a function of the transition stability condition (compare to Supplementary Table S3 and Figure 2 —figure supplement). This indicates that older adults show less signs of a model-based strategy with perfect transition learning in variable-transitions blocks than in stable-transitions blocks. This analysis might suggest that the increased demands imposed by repeated changes in the task transition structure could affect other cognitive mechanism that in turn modulate how effortful model-based decision making is.”

- Descriptive statistics for the parameter estimates for younger and older adults should be reported in a table. At present, the age group effects are reported, but not the estimates themselves.

We now report descriptive statistics for all parameters of both the standard model and the exhaustive model in the supplemental material (Supplementary file 1—table S1 and Supplementary file 1—table S2).